# Modeling and Control of Hysteresis Characteristics of Piezoelectric Micro-Positioning Platform Based on Duhem Model

**Huawei Ji \*, Bo Lv, Hanqi Ding, Fan Yang, Anqi Qi, Xin Wu and Jing Ni**

School of Mechanical Engineering, Hangzhou Dianzi University, Hangzhou 310018, China;
202010084@hdu.edu.cn (B.L.); ll1515735@163.com (H.D.); yangfan1067941909@163.com (F.Y.);
qaq@hdu.edu.cn (A.Q.); wuxin1937@163.com (X.W.); nijing2000@163.com (J.N.)
**\*** Correspondence: jhw76@hdu.edu.cn; Tel.: +86-13185064365

**Abstract:** The hysteresis characteristic of piezoelectric micro-positioning platforms seriously affects its positioning accuracy in precision positioning. It is important to design an effective hysteresis model and control scheme. Based on the analysis of the Duhem model, this paper proposes to divide the hysteresis curve into two parts, the step-up section and the step-down section, to identify the model parameters, respectively, and a hybrid intelligent optimization algorithm based on the artificial fish swarm algorithm and the bat algorithm is proposed. The simulation experiment verified that the error of the improved model was reduced by 48.97%, which greatly improved the identification accuracy of the Duhem model. Finally, an inverse model of the Duhem model for the segmental identification of the improved artificial fish swarm algorithm is established, and a composite controller integrating feedforward, feedback and decoupling control is designed on the basis of the inverse model, and an experimental verification is carried out. The results show that the displacement errors of the composite controller under different voltage signals are all within 0.25%. Therefore, the established model can accurately express the hysteresis characteristics of the platform, and the use of the composite controller can effectively reduce the accuracy error caused by the hysteresis characteristics.

**Keywords:** hysteresis characteristics; Duhem model; segment identification; artificial fish swarm algorithm; composite controller





## 1. Introduction

Precision positioning technology is widely used in microscopes, integrated circuits, biomedical inspections, micro-operation, data storage and other fields, and its rapid development can improve the level of advanced manufacturing technology to a certain extent. Piezoelectric micro-positioning platforms are widely used in precision positioning systems because of their fast response, high resolution, and immunity to magnetic field interference. However, due to the relationship between the characteristics of piezoelectric ceramics, hysteresis nonlinearity will be observed in practical applications, which seriously affects positioning accuracy. Therefore, it is extremely important to accurately model the hysteresis nonlinearity and design a control scheme to improve the positioning accuracy of the piezoelectric micro-positioning platform.

The traditional hysteretic nonlinear models mainly include Preisach model [1,2], Prandtl–Ishlinskii model [3–7], Bouc–Wen model [8,9], neural network model [10–12] and so on. However, the traditional hysteresis nonlinear models cannot show the asymmetry of the hysteresis curve and cannot satisfy the rate dependence of piezoelectric actuators. The Duhem model [13–15] is under constant input, each state is balanced, and the output and input are rate dependent. By adjusting the model parameters, the hysteresis characteristics of the piezoelectric micro-positioning platform under different conditions can be accurately reflected, which meets the requirements of practical applications.

In their research on the parameter identification of the Duhem model, Chen Hui used the recursive least squares method to identify the model parameters; the established Duhem model has high accuracy. Sun Tao used spline curve interpolation and a neural network to identify the parameters of the Duhem model, which effectively improved the identification accuracy of the model parameters. Wang Jingyuan used the least squares method and the gradient correction method to identify the parameters of the Duhem model. The simulation results show that the model built by using the gradient correction method to identify the model parameters is more accurate. It is difficult to describe the asymmetry of the hysteresis curve using the Duhem model obtained by the above parameter identification method; it lacks a practical compensation scheme to improve the positioning accuracy.

M. Eleuteri designed a feedforward compensator based on the inverse Preisach model to describe the hysteresis characteristics of piezoelectric ceramics [16]. Gan proposed a variety of adaptive feedforward controllers using the inverse model of the improved PI model, which eliminated the influence of hysteresis characteristics on the positioning accuracy [17]. Shunli Xiao et al. used a modified Preisach inverse model to compensate the rate-dependent hysteresis nonlinearity of piezoelectric ceramics [18]. However, the feedforward open-loop control based solely on the inverse model relies too much on the design of the model and the accuracy of the identification parameters and cannot eliminate the errors caused by systematic errors and external uncertain factors, and its robustness and stability are very poor.

Aiming at the hysteresis of the piezoelectric micro-positioning platform, segmental parameter identification is carried out on the basis of Duhem hysteresis model modeling, and based on the artificial fish swarm algorithm [19–22], the bat algorithm [23–26] is introduced to improve the identification accuracy of the model parameters. The inverse model of the improved Duhem model is established, and a feedforward controller is designed according to the inverse model for open-loop control, and a PID feedback controller is introduced. Since the research object of this paper is a two-dimensional piezoelectric micro-positioning platform, a decoupling control is designed, a device to reduce the inter-axis coupling rate of the platform. Finally, a composite control integrating feedforward, feedback and decoupling control is realized and the designed composite controller is experimentally verified.

## 2. Research on the Characteristics of Piezoelectric Micro-Positioning Platform

The experimental platform in Figure 1 realizes the identification of model parameters and verifies the validity of the model through the acquisition of real-time data of the piezoelectric micro-positioning platform. The piezoelectric ceramic model is PST150/10/40 VS15; the drive power model is HPV-3C0300A0300; the laser displacement sensor model is LK-G5000; the data acquisition card model is USB-6259 BNC. During the experiment, the computer control system gave a voltage drive signal, and converted it into a continuous analog signal through the data acquisition card, and then amplified the signal through the drive power supply and then loaded it into the piezoelectric micro-positioning platform. At the same time, the laser displacement sensor measured the change in the output displacement in real time and fed the collected displacement to the computer in real time.

In order to study the asymmetry of the hysteresis characteristics, a sinusoidal test voltage signal with a frequency of 1 Hz, a peak value of 150 V, and a bias of 75 V was input to the piezoelectric micro-positioning platform. The test results are shown in Figure 2.

The displacement of piezoelectric ceramics is caused by the electrostrictive effect, inverse piezoelectric effect and ferroelectric effect. Among these, the contribution of the electrostrictive effect to the macroscopic displacement of piezoelectric ceramics is very weak and can be ignored. The output displacement of the inverse piezoelectric effect has a linear relationship with the applied electric field, and there is no hysteresis. The ferroelectric effect displacement mechanism is due to the internal domain inversion of piezoelectric ceramics. When a fixed electric field is applied to the piezoelectric ceramics, the electric domains inside the piezoelectric ceramics will be rotated and elongated to a certain extent along the direction of the electric field, and the boundaries of the domains

will also be elongated and deformed. As can be seen from Figure 2 above, the hysteresis curve is divided into a loading curve and a hysteresis loop. The rising and falling phases of the voltage are not consistent with the output displacement trajectory curve, and as the voltage increases, the hysteresis of the positioning platform becomes more and more obvious. This is because when the applied electric field strength exceeds a certain critical field strength, the piezoelectric ceramic strain is in addition to the inverse piezoelectric effect. The non-180° domain turning starts to dominate (for the piezo crystal strain, only the non-180° domain turning contributes to the displacement of the piezo actuator). During the domain inversion process, there are obstacles inside the crystal that hinder the domain inversion, resulting in energy loss during the domain inversion process. Since this part of the energy loss cannot be recovered, when the field strength decreases, some non-180° domains cannot recover to the same level as when the field strength increases, resulting in the hysteresis of the piezoelectric ceramic. In addition, the greater the field strength, the greater the irreversibility of the non-180° domain turning, and the greater the hysteresis displacement of the piezoelectric ceramic actuator [27].

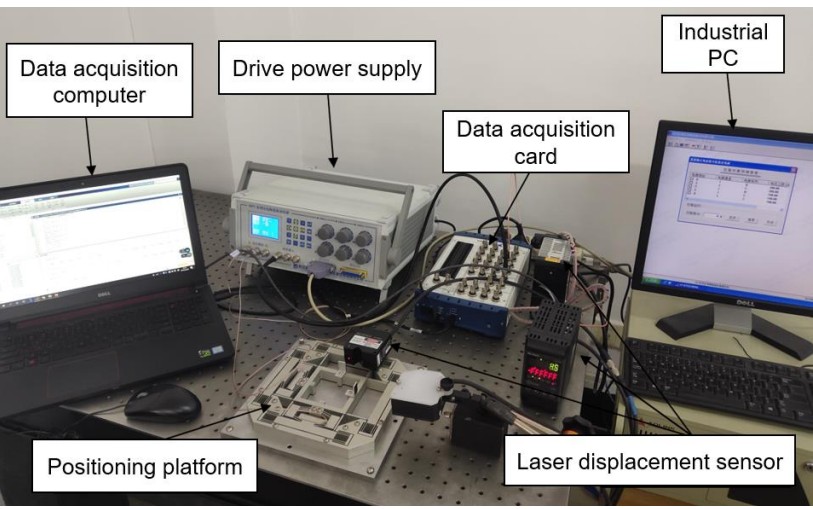

**Figure 1.** Experimental platform.

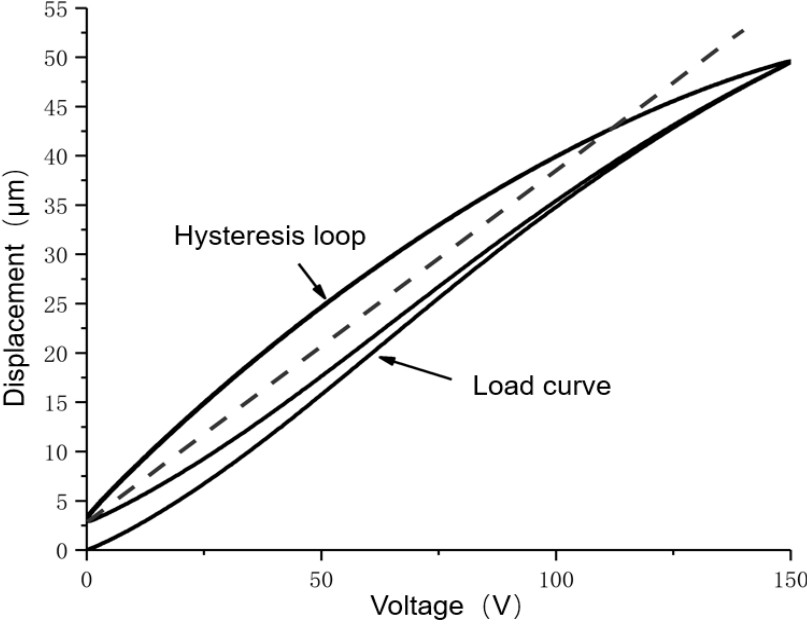

**Figure 2.** Hysteresis graph.

In the rate correlation experiment, sinusoidal voltage signals with frequencies of 1 Hz, 10 Hz, 40 Hz, and 100 Hz were input, and their hysteresis curves were obtained as shown in Figure 3.

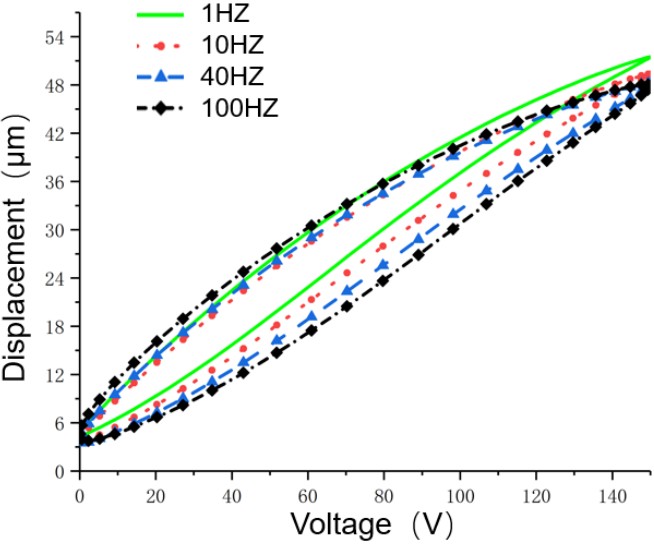

**Figure 3.** Hysteresis graph.

From Figure 3, it can be seen that under different voltage signal frequencies, the hysteresis curves of the positioning platform obviously do not overlap, and as the voltage signal increases, the hysteresis loop becomes wider, and the maximum displacement is also reduced. It also proves the rate dependence of the piezoelectric micro-positioning stage.

The two-dimensional piezoelectric micro-positioning platform studied in this paper has a coupling effect, which will increase the control difficulty of the platform. It is necessary to test the degree of coupling of the platform, define the platform in the $x$ direction and the $y$ direction, and input a sinusoidal voltage signal with an amplitude of 150 V in the y direction. The displacement caused by the coupling effect in the $x$ direction and the output displacement in the $y$ direction can be measured as shown in Figure 4.

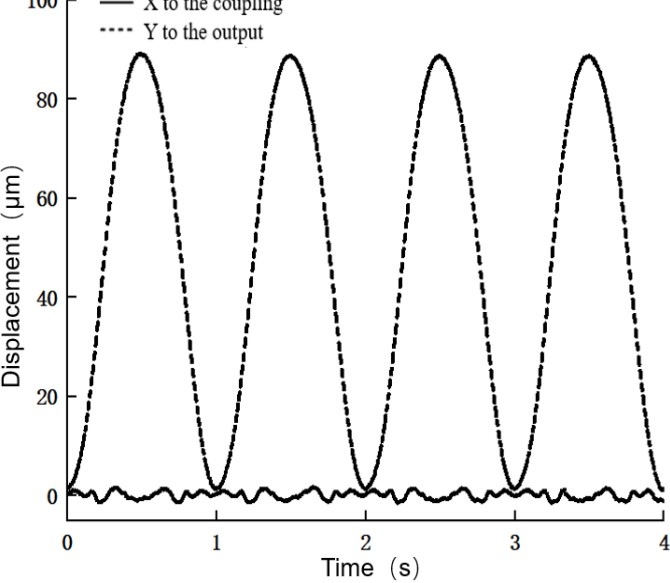

**Figure 4.** Coupling effect.

### 3. Hysteresis Modeling and Parameter Identification

*3.1. Duhem Model*

The basic expression of the Duhem model is:

$$\frac{d_w}{d_t} = \alpha \left| \frac{d_v}{d_t} \right| [f(v) - w] + \frac{d_v}{d_t} g(v) \tag{1}$$

where $v$ represents the hysteresis input of the system; $w$ represents the hysteresis output; and $\alpha$ represents the weight coefficient and satisfies $\alpha > 0$. $f(v)$ and $g(v)$ are two auxiliary functions in the model. By adjusting the parameters $\alpha$, $f(v)$ and $g(v)$, the model can accurately reflect the hysteresis nonlinearity in different situations.

In order to facilitate the subsequent segmental identification of model parameters, the Duhem model expression is rewritten as:

$$\frac{d_{w_{up}}}{d_{v_{up}}} = \alpha \left[ f(v) - w_{up} \right] + g(v), v = 0 - v_{max}^+ \tag{2}$$

$$\frac{d_{w_{down}}}{d_{v_{down}}} = \alpha [f(v) - w_{down}] + g(v), v = v_{max}^+ - 0 \tag{3}$$

where $w_{up}$ represents the output displacement of the boost part of the hysteresis curve; $w_{down}$ represents the output displacement of the buck part; $v_{up}$ represents the input voltage of the boost part; $v_{down}$ represents the input voltage of the buck part; and $v_{max}^+$ represents the maximum positive voltage input by the system.

According to Equations (2) and (3), the expression of the auxiliary function can be obtained as:

$$f(v) = \frac{1}{2\alpha} \left( \frac{d_{w_{up}}}{d_{v_{up}}} - \frac{d_{w_{down}}}{d_{v_{down}}} \right) + \frac{1}{2} \left( w_{up} + w_{down} \right) \tag{4}$$

$$g(v) = \frac{1}{2} \left( \frac{d_{w_{up}}}{d_{v_{up}}} + \frac{d_{w_{down}}}{d_{v_{down}}} \right) + \frac{1}{2}\alpha \left( w_{up} + w_{down} \right) \tag{5}$$

The polynomial approximation of the two auxiliary functions $f(v)$ and $g(v)$ was carried out by using the first approximation theorem of Wellstras, and it was deduced that:

$$w(k) = \begin{cases} \dfrac{w(k-1) + [v(k) - v(k-1)] * \left[ \sum_{i=0}^{n} f_i v(k)^i + \sum_{j=0}^{m} g_j v(k)^j \right]}{1 + \alpha[v(k) - v(k-1)]}, & v(k)j \geq v(k-1) \\[4mm] \dfrac{w(k-1) + [v(k) - v(k-1)] * \left[ \sum_{j=0}^{m} g_j v(k)^j - \sum_{i=0}^{n} f_i v(k)^i \right]}{1 + \alpha[v(k-1) - v(k)]}, & v(k)j < v(k-1) \end{cases} \tag{6}$$

where $v(k)$ represents the voltage input of the system at time $k$; $w(k)$ represents the displacement output of the system at time $k$. It is only necessary to correctly identify the unknowns, $f_i$, $g_j$, and $\alpha$, of the boost and buck stages to establish an accurate Duhem hysteresis model.

*3.2. Segment Identification Model Parameters*

The hysteresis curve is divided into two parts: the step-up section and the step-down section for the segmental identification of model parameters. Based on the artificial fish swarm algorithm, this paper introduces the bat algorithm to optimize it, and uses the optimized artificial fish swarm algorithm to identify the model parameters.

The optimized artificial fish swarm algorithm mainly introduces the bat's acoustic emission frequency $f$ and the global optimal solution $X_{best}$ in the artificial fish swarm algorithm.

After introducing the sonic emission frequency, the speed update expression of the artificial fish can be written as:

$$V_i(t+1) = fV_i(t) + \left( \frac{X_c(t) - X_i(t)}{||X_c(t) - X_i(t)||} \right) * Random() * step \tag{7}$$

where $f$ represents the frequency of the soundwave emitted by the bat, and $V_i(t+1)$ represents the speed of the $i$ artificial fish after iterating $t+1$ times. After introducing the global optimal solution, the speed update expression of the artificial fish can be written as:

$$V_i(t+1) = fV_i(t) + \left( \frac{X_{best}(t) - X_i(t)}{||X_{best}(t) - X_i(t)||} \right) * Random() * step \tag{8}$$

The basic implementation process of the optimized artificial fish swarm algorithm is as follows:

1.  Initialize the algorithm parameters.
2.  The bulletin board is assigned an initial value. Calculate the fitness value of the current position of each artificial fish, record the state $X_{best}$ of the artificial fish in the optimal position and its fitness value to the bulletin board, and judge whether the termination condition is satisfied. If satisfied, go to step 5, if not, go to step 3.
3.  Update the artificial fish position.
4.  Bulletin board information update: Calculate the fitness value of each artificial fish in the new state, compare it with the bulletin board information, and update the bulletin board information if it is better than the bulletin board. It is judged whether the termination condition is met; if so, go to step 5, if not, go to step 3.
5.  The algorithm terminates.

### 3.3. Parameter Identification Results and Analysis

In the model parameter identification, at a certain frequency, the same voltage signal is input, and the experimental value and the theoretical value are compared through simulation, and the fitness function is used for optimization processing. When the experimental value is the closest to the theoretical value, the parameter identification result is output. The objective function chosen in this paper is:

$$J(\theta) = \frac{1}{N} \sqrt{\sum_{k=1}^{N} [w(k) - y(k,\theta)]^2} \tag{9}$$

where $N$ represents the number of sampling points; $w(k)$ represents the actual output displacement of the system at the moment; $y(k,\theta)$ represents the theoretical output displacement of the system at the moment; and $\theta = F(f_0, f_1, \cdots, f_n, \alpha, g_0, g_1, \cdots, g_m)$.

Then, one should apply a sinusoidal voltage signal with a frequency of 1 Hz and an amplitude of 150 V to the positioning platform, and record the output displacement. The Duhem model parameter identification program based on the artificial fish swarm algorithm and improved artificial fish swarm algorithm was developed in MATLAB. Under 1 Hz voltage signal, the accuracy of the whole segment identification and segment identification of the Duhem model parameters by the improved artificial fish swarm algorithm and the accuracy of the segment identification of the Duhem model parameters by the artificial fish swarm algorithm were compared.

The fitting degree of these three models and the experimental data is analyzed. Table 1 and Figures 5 and 6 show the results of the data analysis of the whole identification model of the improved artificial fish swarm algorithm.

It can be seen in Figure 6 that the data of the Duhem model established by the method of identifying the model parameters of the whole segment are poorly fitted with the data measured by the experiment, the maximum error of the output displacement is 3.122 μm, the mean square value of the displacement error of the whole curve reaches 0.784 μm, and the average fitting error is 0.89%.

Table 2 and Figures 7–9 show the results of the data analysis of the improved artificial fish swarm algorithm segmentation identification model.

**Table 1.** Improved artificial fish swarm algorithm whole segment identification model parameters.

| Parameter | Identify the Results |
|---|---|
| $f_0$ | 66.1302 |
| $f_1$ | −17.4632 |
| $f_2$ | 44.4025 |
| $f_3$ | 0.039727 |
| $\alpha$ | 0.0485549 |
| $g_0$ | $5.03691 \times 10^{-4}$ |
| $g_1$ | 18.7986 |
| $g_2$ | −7.53645 |

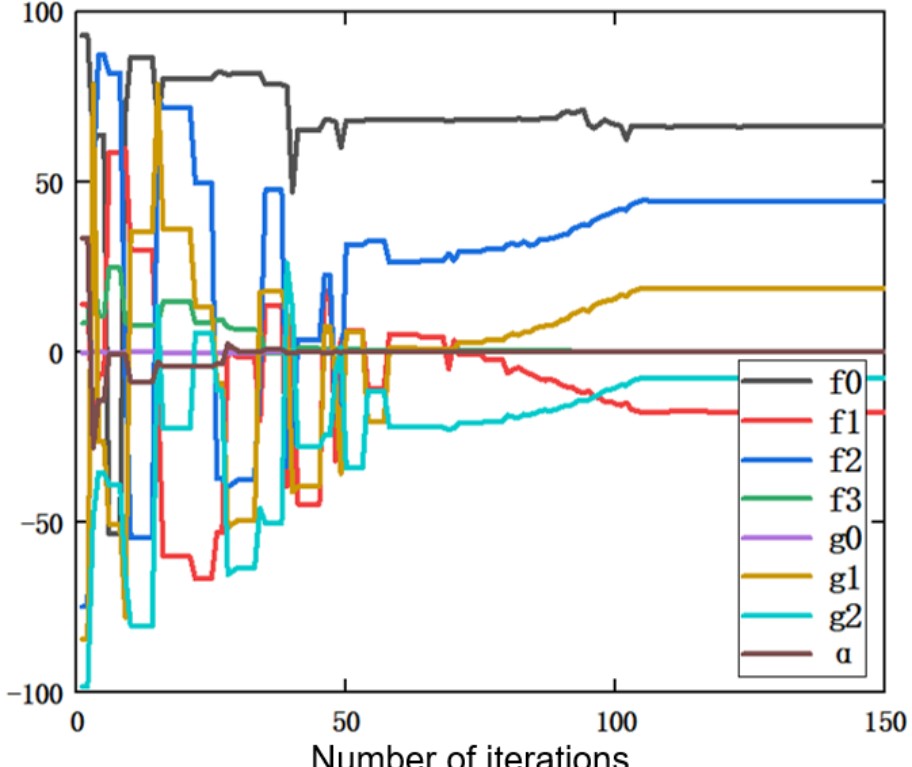

**Figure 5.** Improved parameter convergence process of artificial fish swarm algorithm model.

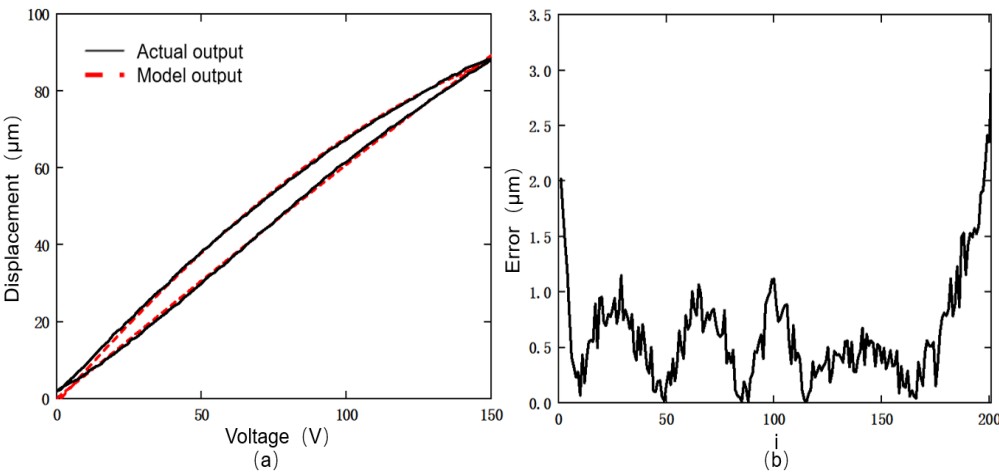

**Figure 6.** Fitting error plot between model data and experimental data: (**a**) fit plot; (**b**) error plot.

**Table 2.** Improved artificial fish swarm algorithm segmentation identification model parameters.

| Parameter | Identify the Results | |
| --- | --- | --- |
| | Rising Segment | Failing Segment |
| $\alpha$ | $-70.5414$ | $65.5115$ |
| $f_0$ | $34.5487$ | $60.9669$ |
| $f_1$ | $6.078683$ | $10.2623$ |
| $f_2$ | $32.7450$ | $32.6779$ |
| $f_3$ | $-70.6247$ | $-65.3047$ |
| $g_0$ | $1.044 \times 10^{-4}$ | $-9.615 \times 10^{-4}$ |
| $g_1$ | $-26.88$ | $-7.82026$ |
| $g_2$ | $2.4939$ | $-11.5234$ |

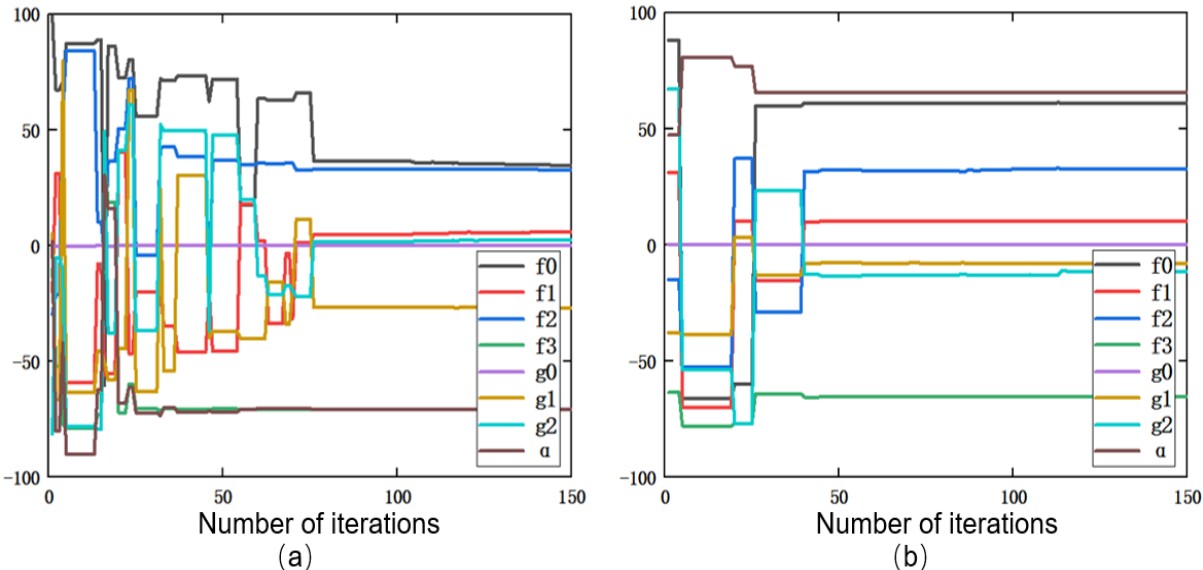

**Figure 7.** Improved parameter convergence process of artificial fish swarm algorithm model: (**a**) ascending segment; (**b**) descending segment.

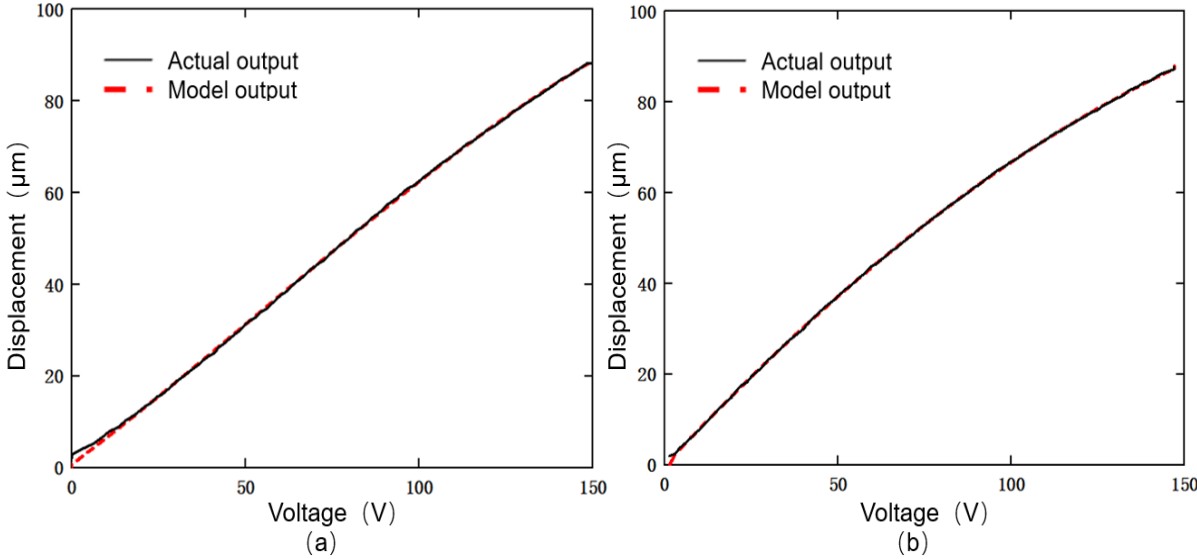

**Figure 8.** Fitting curve between model data and experimental data of improved artificial fish swarm algorithm: (**a**) ascending segment; (**b**) descending segment.

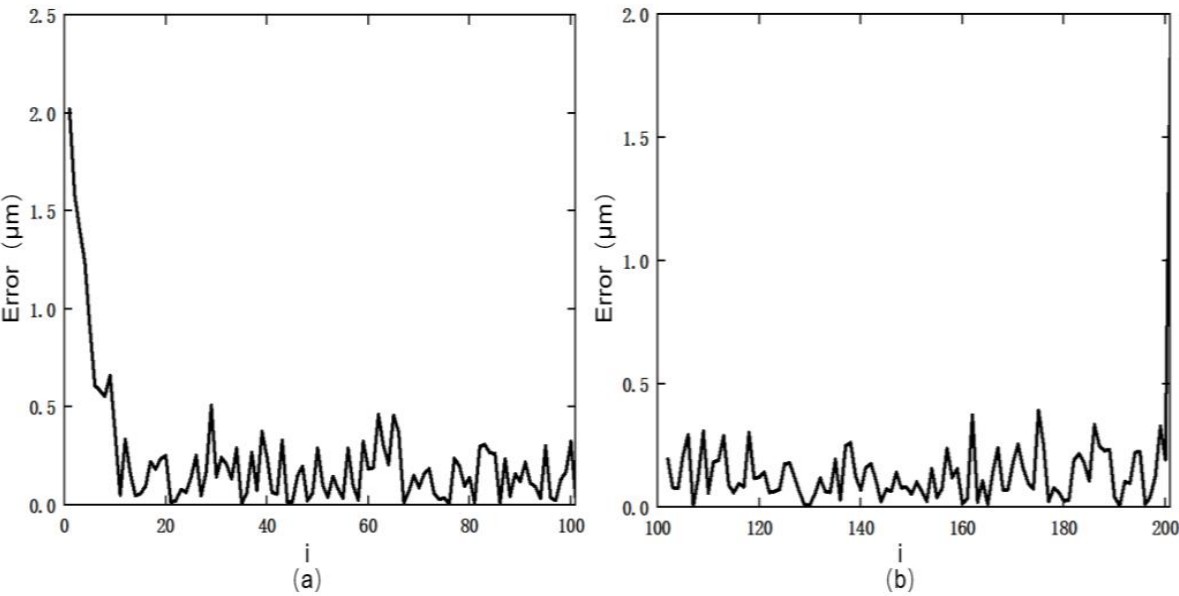

**Figure 9.** Fitting error diagram of model data and experimental data: (**a**) ascending segment; (**b**) descending segment.

It can be seen from Figure 9 that the model data of the rising and falling stages of the segmented Duhem hysteresis model established by the improved method of segmental identification of model parameters are well fitted with the actual data, especially in the depressurization stage; the model output and the actual data are very well fitted. The maximum error between the outputs is only 1.94 μm, and the mean square error of the descending section is 0.1498 μm, while the maximum error between the output displacement and the actual displacement of the ascending section is 2.02 μm, but the mean square error is only 0.3345 μm. The mean square value of the curve error is 0.2385 μm, and the average fitting error rate is 0.27%, which is 69.62% higher than the modeling accuracy of the entire identification model parameters.

Table 3 and Figures 10–12 show the results of the data analysis of the artificial fish swarm algorithm segmentation identification model.

**Table 3.** Artificial fish swarm algorithm segmentation identification model parameters.

| Parameter | Identify the Results | |
|---|---|---|
| | **Rising Segment** | **Failing Segment** |
| $\alpha$ | 20.5173 | 15.5066 |
| $f_0$ | $-38.9701$ | $-43.9303$ |
| $f_1$ | $-20.4347$ | $-52.5417$ |
| $f_2$ | $-9.12629$ | $-9.69358$ |
| $f_3$ | $-20.6836$ | 15.5562 |
| $g_0$ | $8.599 \times 10^{-4}$ | $3.55 \times 10^{-4}$ |
| $g_1$ | 4.4955 | 39.8547 |
| $g_2$ | $-3.00583$ | 18.4739 |

It can be seen from Figure 12 that the maximum error between the output displacement and the actual displacement of the Duhem hysteresis model established by using the artificial fish swarm algorithm to identify the model parameters segmentally is 2.02 μm, and the mean square error is 0.4784 μm; the maximum error of the descending segment is 1.94 μm, and the mean square error is 0.4451 μm; the mean square error of the entire curve is 0.4674 μm, and the average fitting error rate is 0.53%. It can be seen that the modeling error of the improved artificial fish swarm algorithm is reduced by 48.97%.

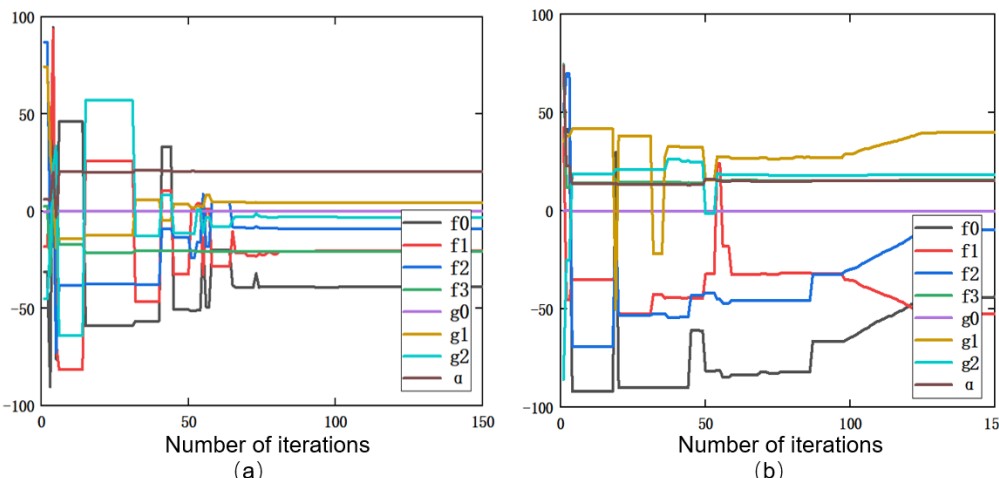

**Figure 10.** Parameter convergence process of artificial fish swarm algorithm model: (**a**) ascending segment; (**b**) descending segment.

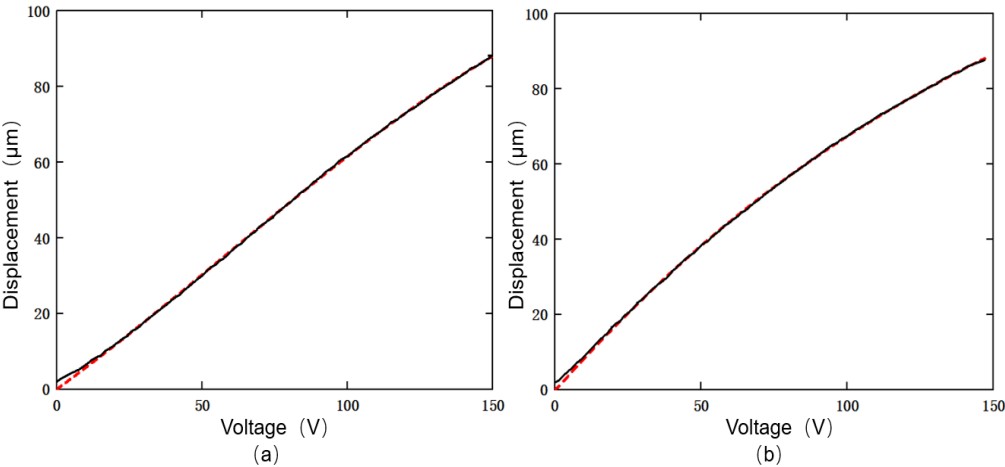

**Figure 11.** Fitting curve between artificial fish swarm algorithm model data and experimental data: (**a**) ascending segment; (**b**) descending segment.

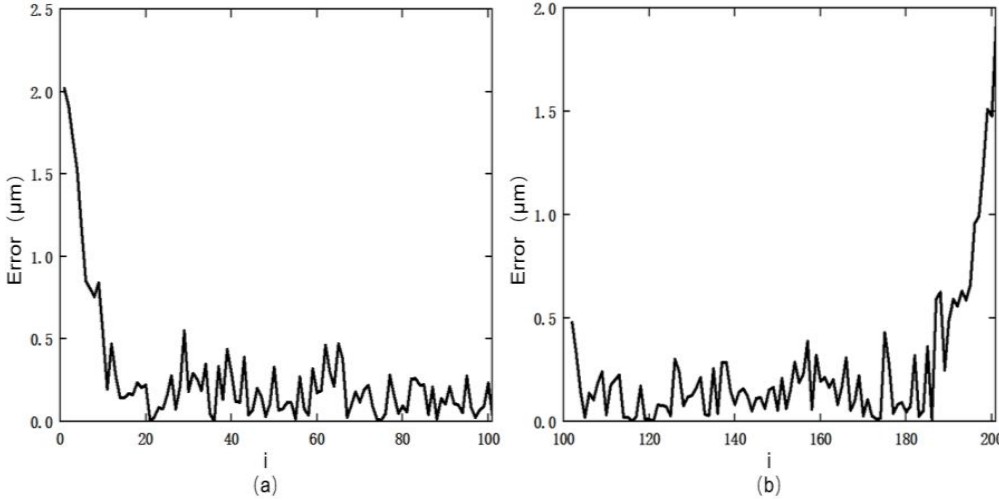

**Figure 12.** Fitting error diagram of model data and experimental data: (**a**) ascending segment; (**b**) descending segment.

### 3.4. Rate Correlation Validation

The accuracy of the model is verified for voltage signals with frequencies of 5 Hz, 10 Hz and 40 Hz.

It can be seen from Figure 13 that under the voltage signal of 5 Hz, the maximum displacement errors of the rising and falling segments of the model are 2.04 µm and 1.87 µm, respectively, and the average fitting error rate is 0.34%.

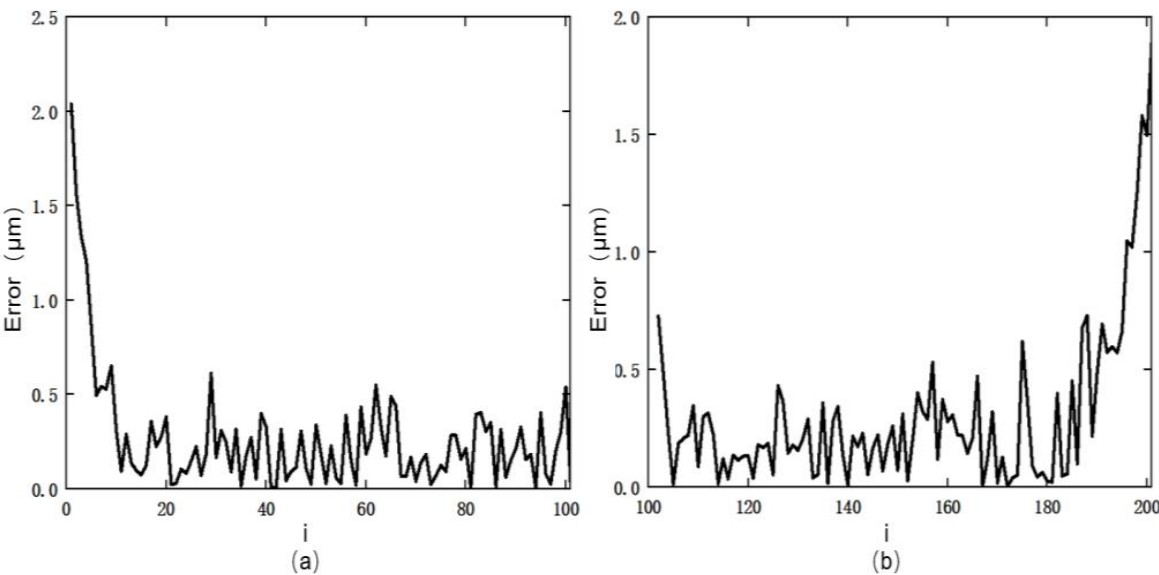

**Figure 13.** Signal fitting error plot (5 Hz): (**a**) ascending segment; (**b**) descending segment.

It can be seen from Figure 14 that under the voltage signal of 10 Hz, the mean square value of the error of the entire curve is 0.3964µm, and the average fitting error rate is 0.45%, which is slightly higher than that of the 5 Hz signal.

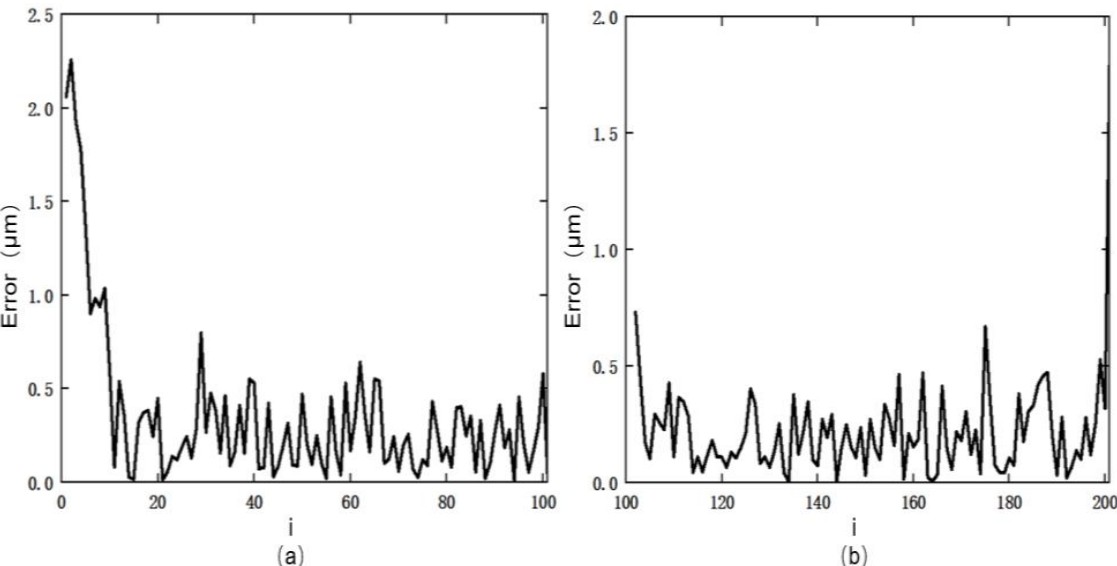

**Figure 14.** Signal fitting error plot (10 Hz): (**a**) ascending segment; (**b**) descending segment.

The simulation results in Figure 15 show that under the voltage signal of 40 Hz, the mean square value of the error of the entire curve is 0.4271 µm, and the average fitting error rate is 0.52%. It can be seen that with the increase in the signal frequency, the accuracy of the model in describing the piezoelectric micro-positioning platform rate correlation decreases slightly, but the built model has high accuracy under voltage signals of different frequencies.

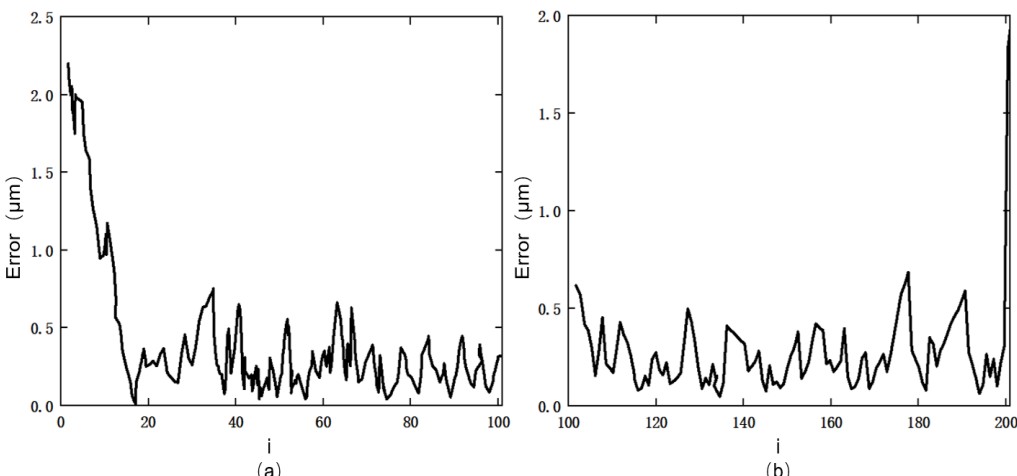

**Figure 15.** Signal fitting error plot (40 Hz): (**a**) ascending segment; (**b**) descending segment.

## 4. Controller Design

### 4.1. Feedforward Controller Design

Before establishing the feedforward controller, it is necessary to invert the built Duhem model. The expression of the inverse model is as follows:

$$\frac{d_{v_{up}}}{d_t} = \frac{\frac{d_{w_{up}}}{d_t}}{\alpha[f(v) - w_{down}] + g(v)}, \quad v_{up} = 0 - v_{max}^+ \tag{10}$$

$$\frac{d_{v_{down}}}{d_t} = \frac{\frac{d_{w_{down}}}{d_t}}{\alpha[f(v) - w_{down}] + g(v)}, \quad v_{down} = v_{max}^+ - 0 \tag{11}$$

The expression of the dynamic discretization inverse model of the Duhem model is further derived as:

$$v(k) = \begin{cases} v(k-1) + \frac{w(k)-w(k-1)}{\left[\sum_{i=0}^n f_i v(k)^i + \sum_{j=0}^m g_j v(k)^j\right] - \alpha w(k)}, & w(k) \geq w(k-1) \\ v(k-1) - \frac{w(k)-w(k-1)}{\left[\sum_{i=0}^n f_i v(k)^i + \sum_{j=0}^m g_j v(k)^j\right] - \alpha w(k)}, & w(k) < w(k-1) \end{cases} \tag{12}$$

where $v(k)$ represents the voltage input of the system at time $k$; $w(k)$ represents the displacement output of the system at time $k$. The schematic diagram of the feedforward controller based on the inverse model is shown in Figure 16.

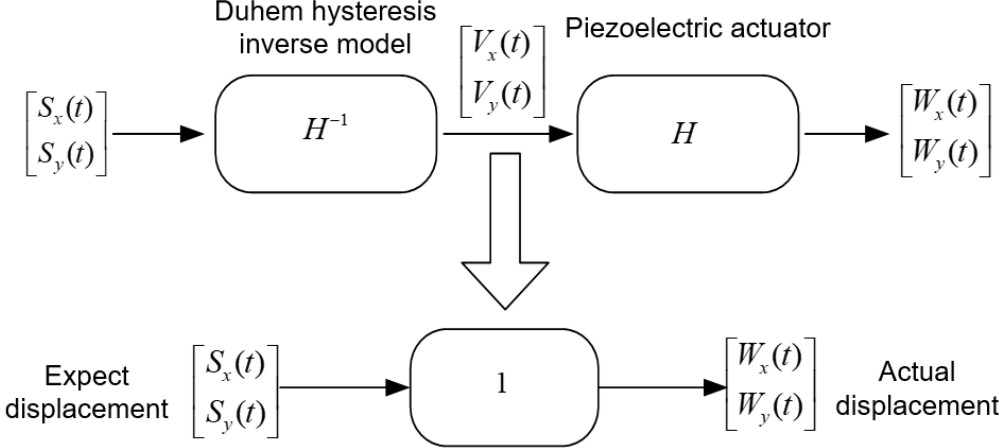

**Figure 16.** Feedforward control schematic.

In the two-dimensional piezoelectric micro-positioning platform, the $x$ direction and the $y$ direction are symmetrical, the hysteresis compensation method only needs to be studied in one direction, and the same compensation method can also be used in the other direction. The output displacement $W_{x(t)}$ in the x direction can be expressed as:

$$W_{x(t)} = H_x\left[V_x(t), V_y(t)\right] \tag{13}$$

where $V_x(t)$ and $V_y(t)$ represent the input voltages in the $x$ direction and $y$ direction, respectively, and $H_x$ represents the output displacement function in the x direction. According to whether it is affected by the coupling effect, $H_x$ can be divided into two types that are only affected by the input voltage in the $x$ direction. The output displacement $H_{xx}$ and coupling displacement $H_{xy}$ are due to coupling effects. Therefore, the feedforward controller in the $x$ direction can be expressed as:

$$W_{x(t)} = H_{xx}\left[V_x(t) \cdot H_{xx}^{-1} V_x(t)\right] + H_{xy}\left[V_x(t), V_y(t)\right] \tag{14}$$

where $H_{xx}^{-1}$ represents the hysteresis inverse model in the $x$ direction, and the feedforward controller expression in the $y$ direction can be obtained in the same way.

### 4.2. Decoupled Controller Design

The basic principle of the $x$ direction decoupling controller is to input a constant voltage signal in the $y$ direction, and no voltage signal is input in the $x$-axis direction. At this time, the displacements generated by the coupling effect are measured, and the voltages required to generate these displacements are calculated. The corresponding voltage signal is applied in the opposite direction of $x$, so as to cancel the coupling displacement generated in the $x$ direction and realize decoupling control. The output displacement expression in the $x$ direction can be written as:

$$W_{x(t)} = H_{xx}\left[V_x(t) + V_{xy}(t)\right] + H_{xy}\left[V_x(t) + V_{xy}(t), V_y(t)\right] \tag{15}$$

$V_{xy}(t)$ represents the voltage used to compensate the coupling displacement in the $x$ direction, and the expression can be written as:

$$V_{xy}(t) = C_{xy} \cdot H_{xx}^{-1}\left[V_x(t) + V_{xy}(t)\right] \cdot H_{xy}\left[V_x(t) + V_{xy}(t), V_y(t)\right] \tag{16}$$

$C_{xy}$ represents the decoupling gain coefficient in the $x$ direction, and similarly, the decoupling controller expression in the $y$ direction can be obtained.

During the decoupling control experiment in the $x$ direction, only a sinusoidal voltage signal with a frequency of 1 Hz and an amplitude of 150 V is input in the $x$ direction. At this point, the output displacement is in the $x$ direction and the coupling displacement is in the $y$ direction. The output displacement after the decoupling controller was added is shown in Figure 17.

Figure 17 reflects the change in the output coupling displacement in the $x$-axis direction before and after the decoupling control. After the decoupling control, the displacement in the $x$-axis direction due to the coupling effect is significantly reduced, the maximum value drops to 0.67 μm, and the mean square error is also reduced to 0.226 μm. It can be seen that the decoupling effect of the decoupling controller in the $x$-axis direction is very obvious, and the mean square value of the coupling displacement before and after the control is reduced by 66.27%.

It can be seen from the experimental results in Figure 18 that the coupling displacement in the $y$-axis direction after control is reduced to 0.652 μm, and the mean square value of the coupling displacement is also reduced from 0.734 μm before the control to 0.215 μm, and the positioning error caused by the coupling before and after the control is reduced by 70.71%. This verifies that the decoupling controller designed in this paper has a good decoupling effect.

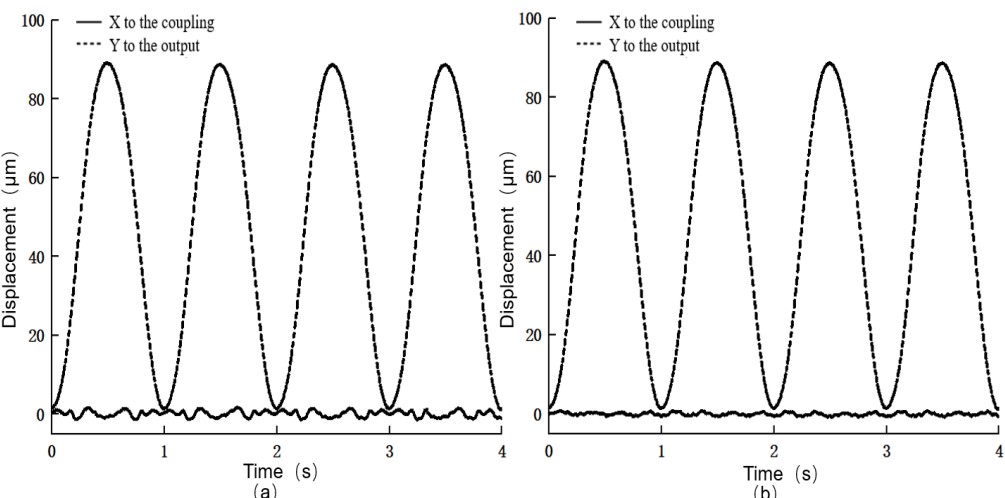

**Figure 17.** Comparison diagram before and after decoupling in the *x* direction: (**a**) before decoupling; (**b**) after decoupling.

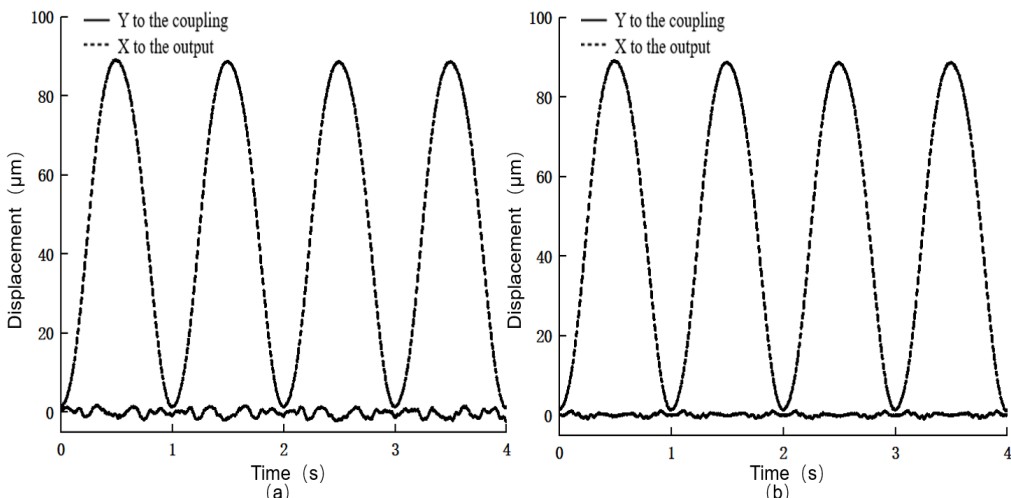

**Figure 18.** Comparison diagram before and after decoupling in the *y* direction: (**a**) before decoupling; (**b**) after decoupling.

*4.3. Composite Controller Design and Testing*

The control accuracy of the feedforward control depends entirely on the accuracy of the inverse model built and whether there are external disturbance factors. In this paper, the feedback controller is introduced to compensate the positioning error, and the decoupling controller is combined to form a composite controller integrating feedforward, feedback and decoupling, which further improves the positioning accuracy of the system.

Introducing the PID controller [28,29] for the feedback control of the system, the composite controller in the *x* direction of the positioning platform can be expressed as:

$$W_x(t) = H_{xx}\big[S_x(t) + S_{xy}(t) - e_x(t)\big] \cdot H_{xx}^{-1}\big[S_x(t) + S_{xy}(t) - e_x(t)\big] + H_{xy}\big[V_x(t) + V_{xy}(t) - V_{ex}(t), V_y(t) + V_{yx}(t) - V_{ey}(t)\big] \quad (17)$$

where $e_x(t)$ is the positioning error in the *x* direction, $S_x(t)$ is the expected displacement in the *x* direction, $W_x(t)$ is the actual output displacement in the *x* direction, and $V_{ex}(t)$ is the voltage value required to generate the positioning error. $e_x(t)$ and $V_{ex}(t)$ are expressed as follows:

$$e_x(t) = S_x(t) - W_x(t) \quad (18)$$

$$V_{ex}(t) = K_{Px}e_x(t) + K_{Ix}\int_o^t e_x(t) + K_{Dx}\frac{de_x(t)}{dt} \quad (19)$$

$K_{Px}$, $K_{Ix}$, *and* $K_{Dx}$ represent the proportional coefficient, integral coefficient and differential coefficient of the feedback controller in the $x$ direction, respectively. This paper uses the trial-and-error method to set these three parameters. In the same way, the composite controller expression in the $y$ direction can be obtained.

In order to compare the control effect of the feedforward controller and the composite controller, the required displacement signal $y(t) = 21 + 24\sin(2\pi tg - \pi/2)$ is input in the piezoelectric micro-positioning stage, where $f$ is 10 Hz and 40 Hz respectively.

The experimental results of the feedforward controller and the composite controller are shown in Figures 19 and 20 below. As can be seen from the figure, the maximum relative error and root mean square error of the composite controller under 10 Hz and 40 Hz signals are significantly reduced compared to the feedforward controller.

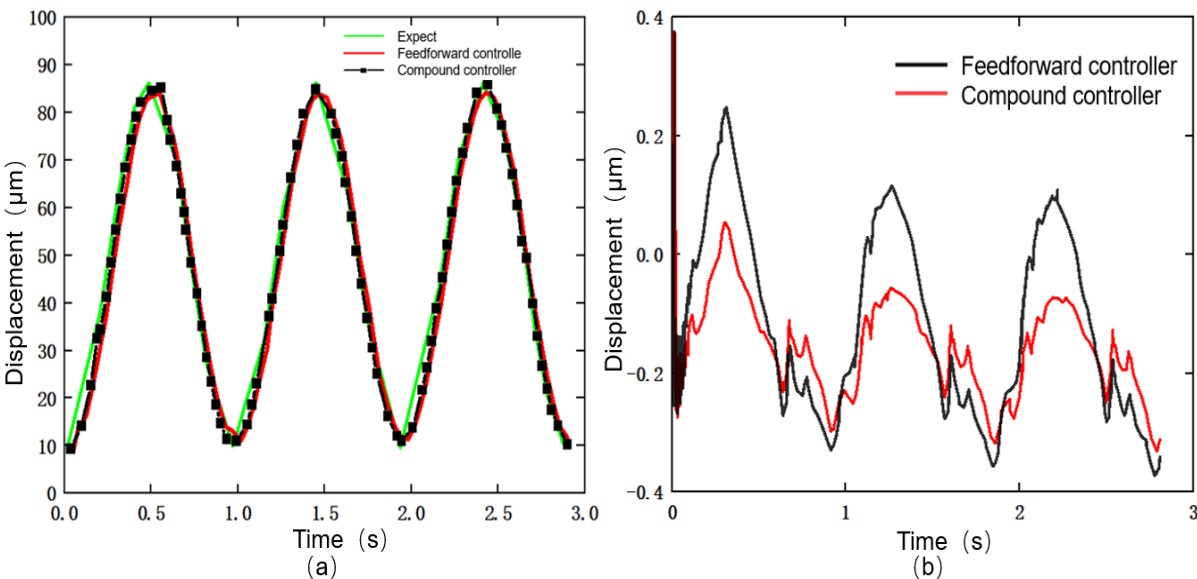

**Figure 19.** Comparison diagram of feedforward control and composite control effect of 10 Hz signal: (**a**) Comparison of expected displacement and actual displacement. (**b**) Error between expected displacement and actual displacement.

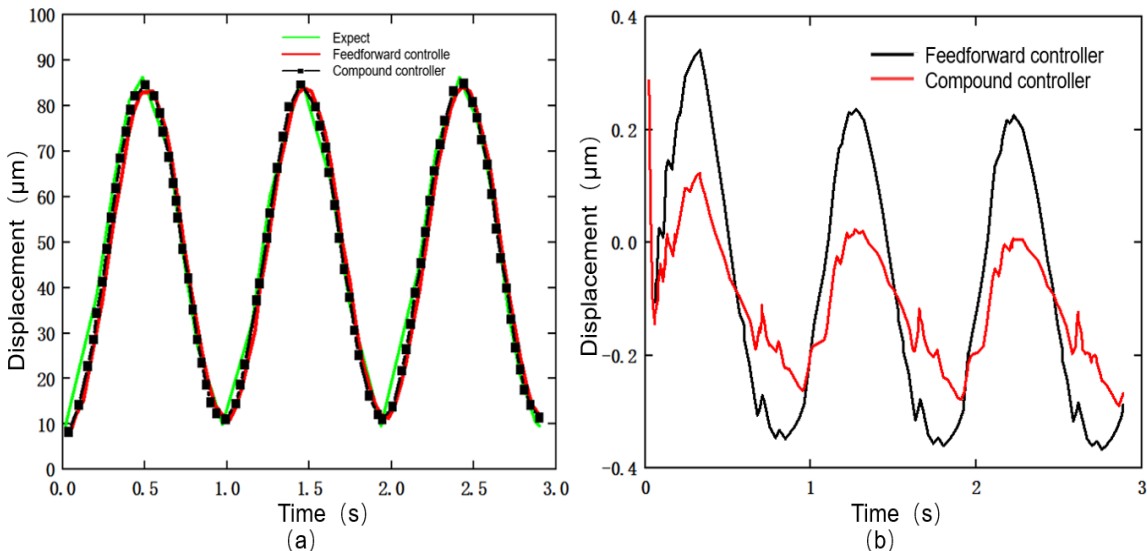

**Figure 20.** Comparison diagram of feedforward control and composite control effect of 40 Hz signal: (**a**) Comparison of expected displacement and actual displacement. (**b**) Error between expected displacement and actual displacement.

External interference is also an important factor affecting the positioning accuracy. In the case of not inputting any voltage signal, displacement detection is performed on one direction of the positioning platform. The experimentally measured displacement error caused by external factors is shown in Figure 21 below; the maximum error is 0.385 μm and the mean square error is 0.1043 μm.

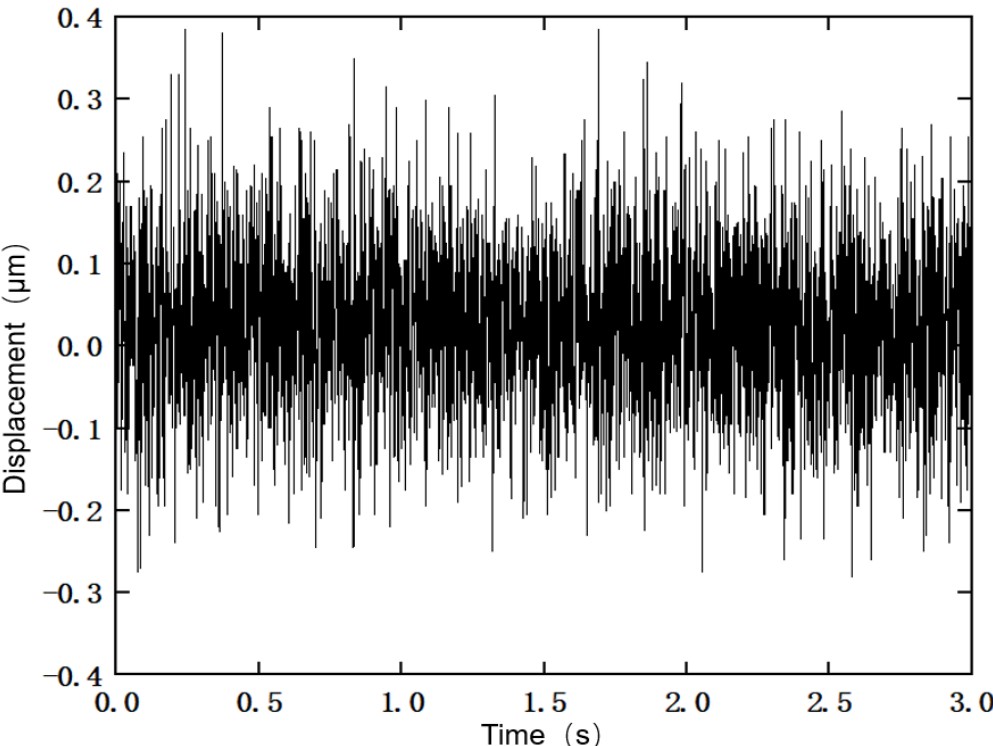

**Figure 21.** Displacement error diagram caused by external factors.

In this paper, the 1 Hz voltage signal was used to carry out a tracking experiment of the composite controller with a constant-amplitude sinusoidal signal and a variable-amplitude sinusoidal signal. In the fixed-amplitude sinusoidal signal tracking experiment, a fixed-amplitude sinusoidal signal with a frequency of 1 Hz and a maximum output displacement of 60 μm was used as the input displacement signal. In the variable-amplitude sinusoidal signal tracking experiment, three cycles of variable-amplitude sinusoidal signals were input into the system as the desired displacement, the signal frequency was always kept at 1 Hz, and the maximum output displacements of each cycle were 60 μm, 40 μm and 20 μm, respectively. Under the two input signals, the displacement tracking errors of the $x$-axis and $y$-axis are detailed below.

It can be seen from the following experimental results that the displacement tracking error range under the constant-amplitude sinusoidal signal is within −0.3608 μm~0.3619 μm, and the average error rate is 0.23%. When the variable-amplitude sinusoidal signal tracking experiment was carried out, the tracking error range was −0.3268 μm and 0.3667 μm, and the average error rate was about 0.213%. It can be seen from Figures 22 and 23 that a one-to-one linearization relationship was basically achieved between the actual output displacement and the expected displacement in the $x$ direction after composite control.

It can be seen from Figures 24 and 25 that the maximum value of the displacement error in the y direction under the composite control was 0.3738 μm in the constant-amplitude sinusoidal signal, and the average error rate remained at 0.236%. In the variable-amplitude sinusoidal signal tracking experiment, the tracking error is between −0.3197 μm and 0.3842 μm, and the average error rate is 0.215%. The experimental results fully prove that the composite controller designed in this paper can effectively improve the positioning accuracy of the positioning platform.

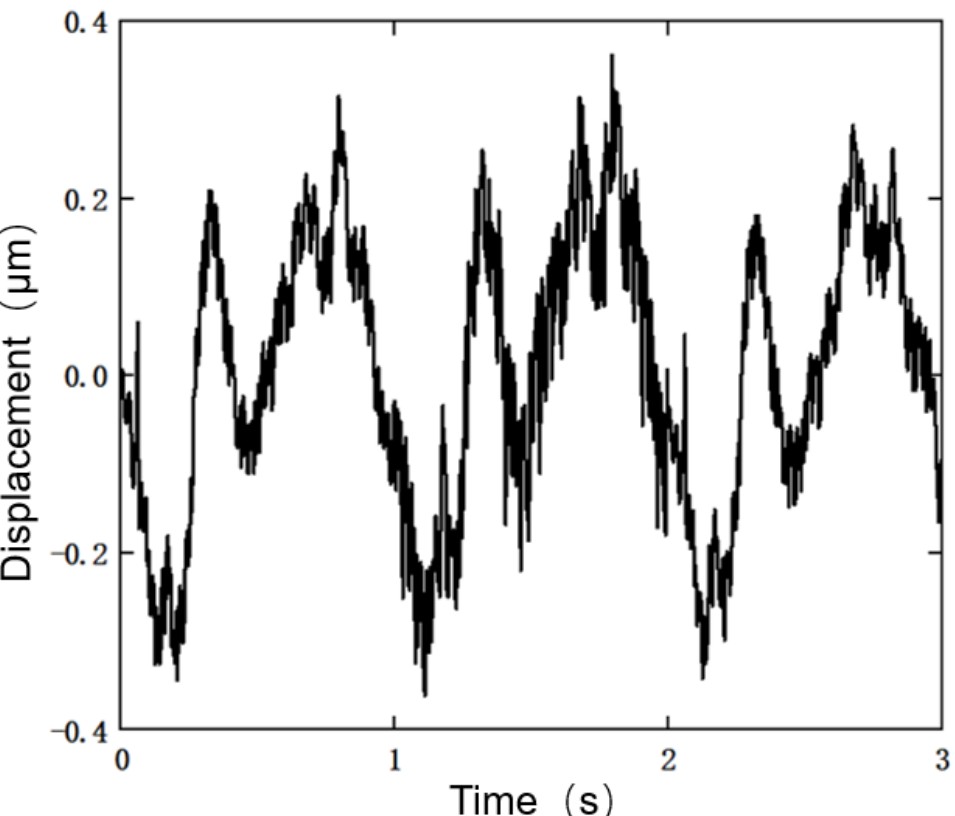

**Figure 22.** Displacement error diagram of *x* direction constant-amplitude signal.

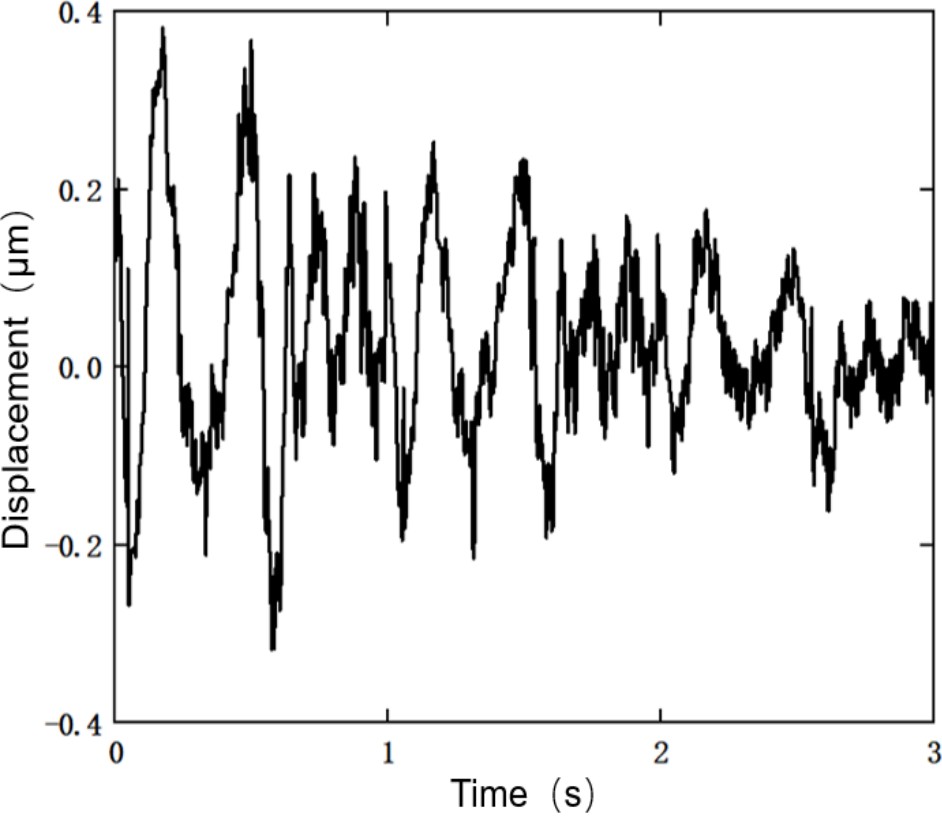

**Figure 23.** Displacement error diagram of *x* direction variable-amplitude signal.

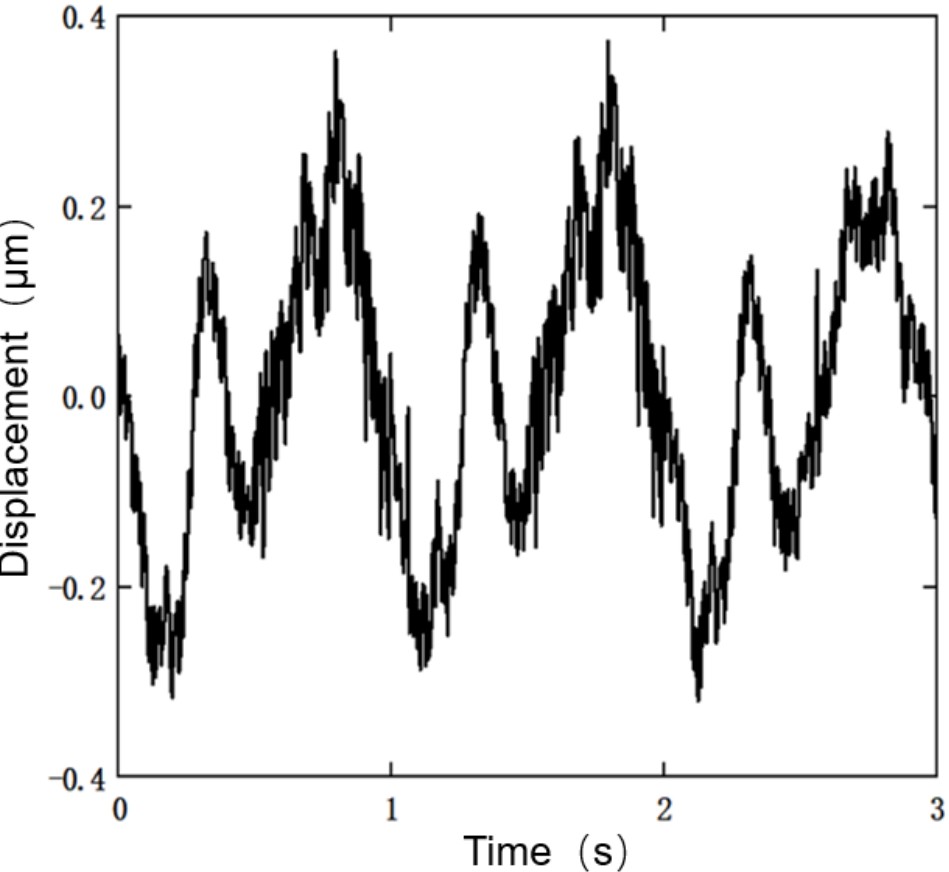

**Figure 24.** Displacement error diagram of *y* direction constant-amplitude signal.

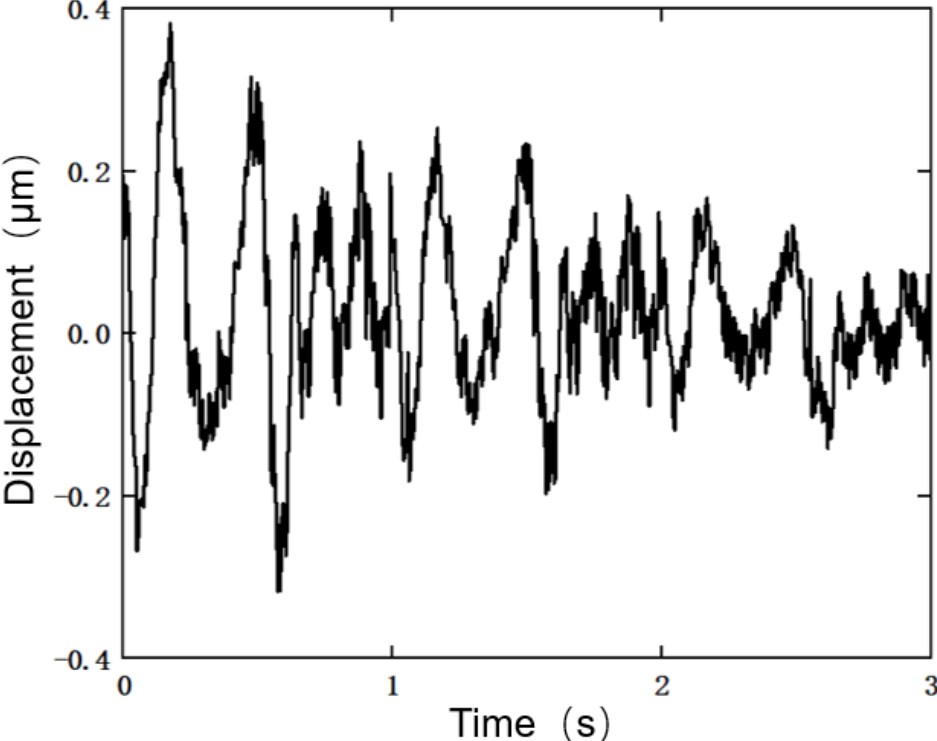

**Figure 25.** Displacement error diagram of *y* direction variable-amplitude signal.

## 5. Conclusions

In this paper, the two-dimensional piezoelectric micro-positioning platform is taken as the research object. In order to reduce the positioning error caused by the hysteresis characteristics of the positioning platform, a Duhem hysteresis model is established. The hysteresis curve is divided into two parts: boost section and step-down section for model parameter identification, thereby establishing a segmented Duhem model. Additionally, based on the artificial fish swarm algorithm, the bat algorithm is introduced to optimize the model. The inverse model of the established model is established, and on this basis, a composite controller integrating feedforward, decoupling and feedback control is designed. The main research results are as follows:

(1) The hysteresis curve was divided into the step-up section and the step-down section for model parameter identification. The segmented Duhem model established from this can more accurately describe the hysteresis characteristics of the positioning platform, and the modeling accuracy was improved by 69.62%.

(2) After introducing the bat algorithm to optimize the artificial fish swarm algorithm, the identification accuracy of the model parameters greatly improved, and the modeling error was reduced by 48.97%.

(3) The composite controller designed based on the established Duhem inverse model, which integrates feedforward, decoupling and feedback control, has displacement errors under both constant and variable-amplitude sinusoidal signals within 0.25%.

**Author Contributions:** Conceptualization, H.J., B.L., H.D., F.Y., A.Q., X.W. and J.N.; methodology, H.J., B.L. and H.D.; validation, H.J.; investigation, H.J. and B.L.; data curation, F.Y., A.Q., X.W. and J.N.; writing—original draft preparation, H.J.; writing—review and editing, H.J. and B.L.; visualization, H.J.; supervision, F.Y., A.Q., X.W. and J.N. All authors have read and agreed to the published version of the manuscript.

**Funding:** This research received no external funding.

**Institutional Review Board Statement:** Not applicable.

**Informed Consent Statement:** Not applicable.

**Data Availability Statement:** The data presented in this study are available on request from the corresponding author. The data are not publicly available due to privacy.

**Conflicts of Interest:** The authors declare no conflict of interest.

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
