# Peer review of "Modeling and Control of Hysteresis Characteristics of Piezoelectric Micro-Positioning Platform Based on Duhem Model"

_actuators, doi:10.3390/act11050122_

Round 1

Reviewer 1 Report

A paper entitled "Modeling and Control of Hysteresis Characteristics of Piezoelectric Micro-positioning Platform Based on Duhem Model " studies the parameter identification method of Duhem model, and designs a compound controller integrating feedforward, feedback and decoupling control. The effectiveness of the proposed model parameter identification method and control scheme is verified by experiments. Overall, the paper is in good quality and could be published after adding the following:

1) In Section 3.3, in addition to the error graph, the author would preferably give a comparison graph of the model output and the actual output of the piezoelectric micro-positioning platform.

2) In Section 3.4, in addition to the error plot, the authors should add a comparison plot of the model output at higher frequencies (eg, 40 Hz) and the actual output of the piezoelectric micro-positioning platform.

3) The proposed Duhem model has a certain rate dependence, and a feedforward controller is designed on the basis of its inverse model. The authors should supplement the tracking effects of the feedforward controller and the composite controller under 10 Hz and 40 Hz displacement signals.

Reviewer 3 Report

Precision positioning technology has a wide range of practical applications, however, piezoelectric micro-positioning platforms, which are widely used in practice, exhibit hysteresis nonlinearity, which affects their positioning accuracy, so the issue of improving positioning accuracy is very important. Therefore, the present article is relevant and interesting, and the results obtained by the authors contribute to the further practical development of this technology. My comments are listed below.

1) It would be better to add a description of the experimental platform to the text of the manuscript, and not only show it in Figure 1. The same comment goes for Tables 2 and 3, as well as Figures 7 and 9 - they should be explained in the text of the manuscript.

2) It would be better to increase the space between the text and Figures 3 and 14 as they almost overlap the text.

3) In addition, careful proofreading of the text is required. For example, in line 77 there is an unnecessary dot, additionally, between the value of the quantity and the units of measurement, as a rule, it is necessary to put a space, and so on.

Round 2
